

# Integrated transcriptome and metabolome revealed the drought responsive metabolic pathways in Oriental Lily (*Lilium L.*)

Zhenkui Cui[1], Huaming Huang[1], Tianqing Du[2], Jianfeng Chen[1], Shuyan Huang[1] and Qushun Dai[1]

[1] Department of Landscape Architecture, Fujian Forestry Vocational & Technical College, Nanping, Fujian, China
[2] College of Agriculture, Shanxi Agricultural University, Taigu, Shanxi, China

## ABSTRACT

**Objective:** Lily is an essential ornamental flowering species worldwide. Drought stress is a major constraint affecting the morphology and physiology and lily leaves and flowers. Therefore, understanding the molecular mechanism underlying lily response to drought stress is important.

**Method:** Transcriptome and metabolome analysis were performed on Oriental Lily subjected to drought stress.

**Result:** Most transcription factors and metabolites yielded by the conjoint analysis displayed a downregulated expression pattern. Differential genes and metabolites mainly co-enriched in glycolic pathways related to sugars, such as galactose, and sucrose, glycolysis and gluconeogenesis, indicating that drought stress reduced the sugar metabolism level of Oriental Lily. Combined with transcriptome and metabolome data, nine pairs of differentially expressed metabolites and the genes ($p < 0.05$) were obtained. Interestingly, a gene named *TRINITY_DN2608* (encoding a type of alpha-D-glucose) cloned and its overexpression lines in *Arabidopsis thaliana* was generated. Overexpression of *TRINITY_DN2608* gene elevated the susceptibility to drought stress possibly by suppressing the glucose level.

**Conclusion:** The enrichment of sugar-related pathways advocates the potential role of glucose metabolism in drought stress. Our study provides theoretical information related to the glucose-mediated drought response and would be fruitful in future lily breeding programs.

# INTRODUCTION

Lilium is a perennial herbaceous bulb plant of genus Lilium in the Liliaceae family, *lily* ranked in the top five cut flower providing aesthetic and medicinal values (*Jin et al., 2020*; *Kong et al., 2021*). The underground bulbs of lily plants are directly affected by the availability of water. For instance, excess water in the rhizosphere can easily cause bulb rot whereas its scarcity could lead to crippled bulb growth (*Wang et al., 2020*; *Yan et al., 2022*). Low soil moisture significantly hampered the lily plant height and underground parts thus cutting down the overall yield (*Shahba, 2019*; *Li et al., 2020*). More particularly, drought

Corresponding author
Zhenkui Cui, cuifang_gao@126.com

stress altered the leaf morphology, number of leaves and abating the thickness of spongy tissue in lily plants (*Hura, Hura & Ostrowska, 2022*). Therefore, identifying the drought responsive genes and metabolites is of great importance to minimize the losses in yield of lily plant (*Hura, Hura & Ostrowska, 2022*). A series of articles stressed the detrimental effects of prolonged drought stress over plant height, root system and flower development (*Cui et al., 2016*). The negative influence of drought stress on lily is in part attributed to decreased chlorophyll content and stimulation in electrical conductivity (*Li et al., 2023*). In addition, under drought stress, the contents of malondialdehyde (MDA), soluble sugar and soluble protein of lilium increased, and the activity of antioxidant enzymes decreased to different degrees (*Lastdrager, Hanson & Smeekens, 2014*). In cotton plant, drought stress can damage the fiber yield and quality by altering the expression of several drought responsive genes (*Zheng et al., 2022*; *Duan et al., 2023*). ROS homeostasis is crucial for maintaining the cell turgidity and plasticity of wheat plant under drought stress (*Tian et al., 2023*). However, the underlying molecular mechanism regulating lily response to drought stress is elusive.

The rapid advancement in biotechnology has enabled plant researchers to employ transcriptomics and metabolomics tools. The utility of transcriptome and metabolome analyses of 'Hequ Red millet' (HQ) and 'Yanshu No.10' (YS10) millet revealed several pathways, including starch and sucrose metabolism, pyruvate metabolism, metabolic pathways, and the biosynthesis of secondary metabolites (*Cao et al., 2022*). *Tamarix taklamakanensis* is a perennial shrub possessing generally drought tolerant behavior that was used for metabolome and transcriptome analysis. The result revealed a large number of DEGs enriched in tryptophan and alpha-linolenic acid metabolism, flavonoid and phenylpropanoid biosynthesis and the mitogen-activated protein kinase (MAPK) signaling pathway (*Sun et al., 2022*). *Zanthoxylum bungeanum Maxim*. leaves (ZBLs) are rich in flavonoids, becoming a popular food in its agroecological region. *Hu et al. (2022)* performed integrated analysis of transcriptome and metabolome data to uncover the regulatory mechanism of flavonoid components under drought stress. Despite the fruitful results, application of conjoint metabolome and transcriptome analysis in drought-stressed lily is lacking.

In this study, Oriental Lily was selected as the material, and transcriptome and metabolome analysis were used. Bionfirmatics tools were utilized to mine highly enriched pathways and drought-responsive genes. Further, functional verification of candidate gene, was also performed which could be beneficial in future lily research programs.

## MATERIAL AND METHOD

### Test materials and drought treatment

Oriental Lily was planted in the garden laboratory of Fujian Forestry Vocational Technical College. The bulbs (average mass 65 g) were planted in plastic pots (15 cm (H) × 10 cm (inner diameter)), containing three small holes in the pelvic floor to ensure ventilation of the underlying soil. Each pot contained only one seed ball. The volume ratio of cultivated substrate was 3:1 (coconut brick and nutrient soil, pH = 6.15). Each pot contained approximately 350 g substrate.

A total of 60 lily seedlings with basically the same growth and no pests and diseases were selected. After 7 days of culture, they were placed in the artificial climate box to slow the seedlings. The ambient conditions were set as: day temperature (25 ± 1) °C, night temperature (18 ± 1) °C, light intensity 12,000 Lx, photocycle 12 h/12 h (day/night), relative humidity 60%.

The experiment set up two water treatments, each treatment had five replicates: the control group had normal water supply (ZC), and the soil moisture content was 70–80% of the field water holding capacity; in the drought stress group (GH), the water control was 40–50% of the field water holding capacity. Water loss was measured daily at 5 p.m. by weighing and water control, while portable TDR (TZS-PHW, China) was used to measure soil volumetric water content.

## Paraffin section preparation

The fresh tissue was fixed with fixed liquid for more than 24 h. We removed the tissue from the fixed liquid and trimmed the target tissue with a scalpel in the ventilation cupboard and then transferred to dehydration box. The dehydration box was shifted to the dehydrator in order to dehydrate with gradient alcohol. The rest of the procedure was performed following the method of *Yuan et al. (2023)*.

## Observation of morphological indicators and observation of leaf anatomy

The growth indexes of each lily plant were recorded at 10, 15, 20, 25 and 30 days after treatment, and the leaves of the same leaf position were measured. The specific morphological indexes include leaf length, leaf width and leaf area using a centimeter-scale ruler. The thickness of blade, palisade tissue and spongy tissue were measured by Motic Image Plus 2.0 measuring software, and each structural parameter was the average value measured within 20 field of view.

$$CTR = \frac{\text{Thickness of fence tissue}}{\textit{Thickness of leaf}} \times 100\% \tag{1}$$

$$SR = \frac{\text{Thickness of spongy tissue}}{\textit{Thickness of leaf}} \times 100\%. \tag{2}$$

## Sample preparation for metabolome analysis

The vacuum freeze-dried method was used for biological samples to grind the samples (MM 400, Retsch); 100 mg of powder was weighed and dissolved in 1.0 mL of extract and refrigerated overnight at 4 °C with vertexing 3 times. The sample underwent centrifugation for 10 min at 12,000 rpm, the supernatant was aspirated and filtered with a 0.22 μm microporous membrane for liquid chromatography tandem mass spectrometry (LC-MS/MS) analysis.

## Data processing and multivariate analysis

The Proteowizard software package was used to convert the original mass spectrometry file into mzXML file format (*Smith et al., 2006*), and the RXCMS software package was used to analyze the chromatographic peaks. The obtained qualitative and quantitative results of metabolites. *Dos Santos & Canuto (2023)* were saved for further analysis. The identified metabolites were annotated using HMDB (*Wishart et al., 2022*), massbank (*Horai et al., 2010*), LipidMaps (*Sud et al., 2007*), mzcloud (*Abdelrazig et al., 2020*) and KEGG databases, (*Kanehisa et al., 2022*). The parameter was set to ppm <30 ppm (*Asif et al., 2023*).

The R software package Ropls was used for principal component analysis of normalized data (*Thevenot et al., 2015*). On the basis of orthogonal partial least squares discriminant analysis (OPLS-DA), score plot, load plot and S-plot were plotted to show the difference of metabolite composition among samples. The *p* value was calculated according to the statistical test, the projected importance of variable (VIP) was calculated by OPLS-DA dimensionality reduction method, and the component difference multiple was calculated by fold change. The differential metabolites were identified with $p < 0.05$ and VIP > 1 as the screening conditions for differential compounds.

## Pathway analysis

The Metabolomic Analyst software package was used to conduct functional pathway enrichment and topological analysis of the screened differential metabolites.

The enrichment results were taken as the unit of KEGG pathway, and hypergeometric test was applied to obtain the *p*-value of pathway enrichment. With *p*-value ≤ 0.05 as the threshold.

$$p\text{-value} = \sum_{i=m}^{M} \frac{\binom{M}{i}\binom{N-M}{n-i}}{\binom{N}{n}} = 1 - \sum_{i=0}^{m-1} \frac{\binom{M}{i}\binom{N-M}{n-i}}{\binom{N}{n}}. \tag{3}$$

## Library construction and sequencing for transcriptome analysis

Total RNA was extracted, and the Oligo (dT) magnetic beads were used to enrich the mRNA with polyA structure. Using RNA as a template, the first strand of cDNA was synthesized using 6-base random primers and reverse transcriptase, and the second strand cDNA was synthesized using the first strand cDNA as template. Then, the constructed library was inspected by Agilent 2100 Bioanalyzer (Agilent, Santa Clara, CA, USA), and the total concentration and effective concentration were detected. The libraries were put forward for paired-end (PE) sequencing using next-generation sequencing (NGS) based on Illumina sequencing platform (*Eid et al., 2009*). The longest transcript was selected as Unigene, and further used for GO, KEGG, eggNOG, SwissProt, Pfam annotation, ORF prediction, SSR prediction (*Su et al., 2021*).

## Quantitative real-time PCR

Total RNA was extracted using the RNA Easy Fast Plant Tissue Kit (Tiangen, Beijing, China) according to the manufacturer's instructions. The expression were analyzed by qRT-PCR using the SYBR green PCR mix (Vazyme Biotech Co., Ltd, Beijing, China). The primer details are available for *TRINITY_DN2608* (F:TGAACACCTCT TGTCGAGCCR:CCGTCAACCTGACCGTAACA). Actin expression was used as an internal control. (F:ATTCAGATGCCCAGAAGTCTTGTTR:GAAACATTTTCTGT GAACGATTCCT). The relative gene expression was calculated by $2^{-\Delta CT}$ (*Livak & Schmittgen, 2001*).

## Arabidopsis transformation

The open reading frame of the cloned *TRINITY_ DN2608* gene was inserted into the plant expression vector pCambia 1305. The restriction enzymes used in this experiment are *EcoRI* and *NotI*. Agrobacterium vectors were inoculated into LB liquid medium containing 25 mg·L$^{-1}$Rif and 50 mg·L$^{-1}$Kana resistance for activation culture for 2 days. The culture conditions were set as 28 °C and 200 rpm. According to the ratio of bacteria solution: LB medium = 1:50 (V/V), that is, 1 mL of the above bacteria solution was absorbed and added into LB liquid medium containing the corresponding resistance at 28 °C, overnight culture at 200 rpm until the OD600 is between 0.6–1.0. On the second day, the mixed transformation vector was placed in a 50 mL centrifuge tube, and the inflorescences of the white flower buds were immersed in the bacterial solution for about 30 S. After transformation, the plants were shaded with black plastic bags and placed in a article box for 24 h of moisture. At the end of shading, the material was placed under normal conditions for growth, and the inflorescences were re-treated with the above transformation method after 1 week. Finally, the material was placed under normal conditions to grow until the seeds matured (*Jia et al., 2023*).

## RESULTS

### Phenotypic analysis of lily under drought stress

The drought-stressed lily plants significantly restricted the growth compared to that of control (CK) (Figs. 1A and 1B). Under drought stress, the length, width and area of lily leaves were measured (Figs. 1C–1F). A significant decrease in the length and width was observed following drought stress. Noticeable difference in the length and width of CK and drought-stressed lily plants was recorded at the 15th day. For the leaf area, the significant difference between CK and drought-stressed lily plants was observed at the 20th day. Paraffin section experiment was used to investigate the effect of drought stress on the anatomical structure of lily leaf (Figs. 1G–1J). That thickness of fence tissue was about 54.49 μm under drought stress, which was approximately 12.87% lower than the control. The spongy tissue with a thickness value of 104.55 μm under drought stress was 11.41% lower than the control. Meanwhile, the drought-stressed leaf thickness was recorded at a value of 161.14 μm, which was 10.05% lower than the control. Following the morphological response of lily to drought stress, samples were taken from the control and drought treated plants for transcriptomic and metabolomic analysis.

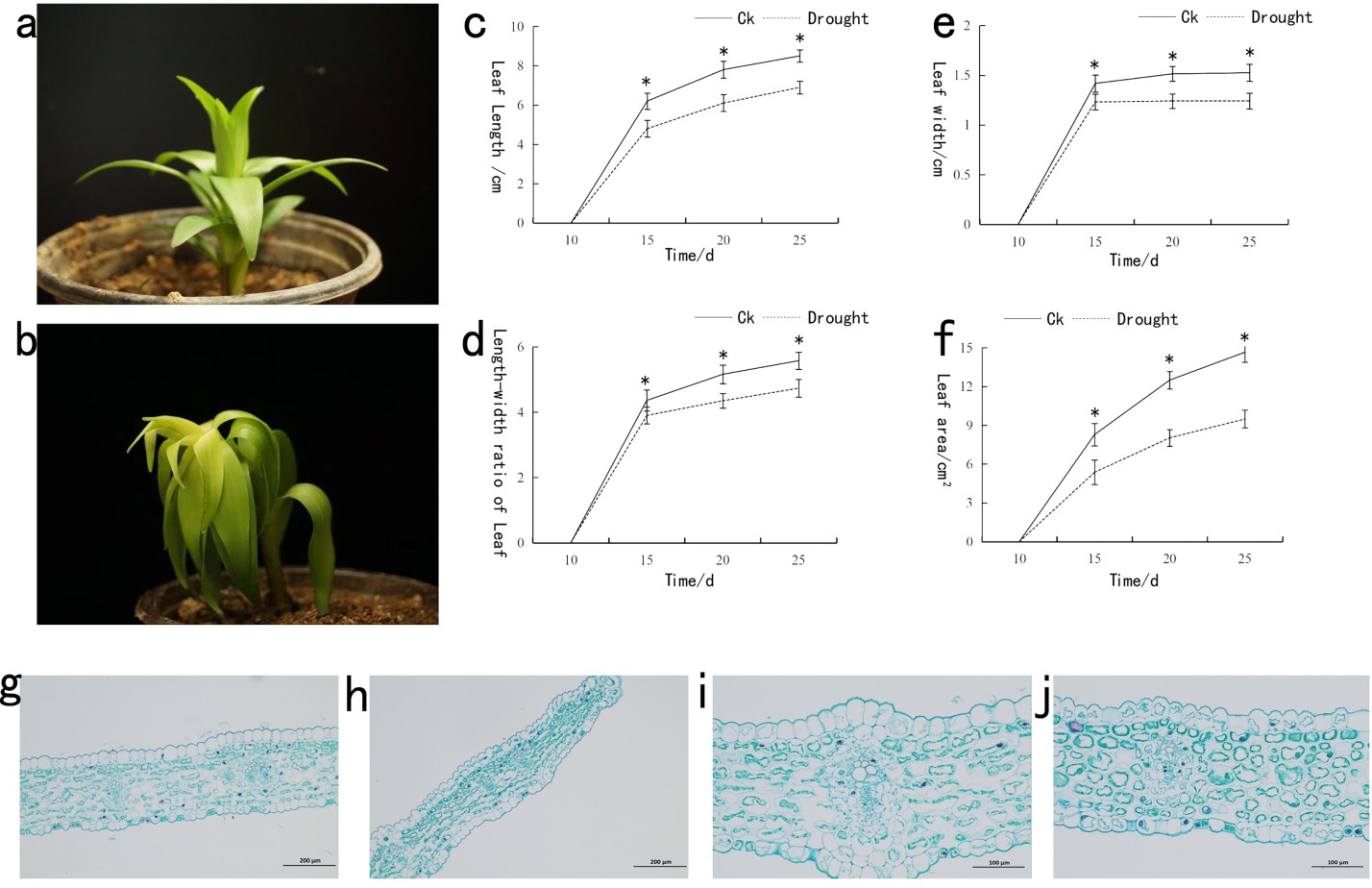

**Figure 1** **Phenotypic analysis of lily under drought stress.** (A) Represents the phenotype of lily under normal, (B) after drought stress, (C) leaf length before and after drought stress, (D) leaf width before and after drought stress, (E) ratio of length/width of lily before and after drought stress, (F) area of lily leaves before and after drought stress, (G) 100x mirror leaf anatomical structure under normal, (H) 100x mirror leaf anatomical structure after drought stress, (I) 200x mirror leaf anatomical structure under normal, (J) 200x mirror leaf anatomical structure after drought stress. The asterisk (*) in the figure is on the normal plant line, indicating a significant difference at the 0.01 level.

## Metabolomic analysis of Oriental Lily

The primary screening yielded a total of 37,285 metabolites, among which 3,057 were down-regulated and only 688 were up-regulated, while the remaining were insignificant (Fig. 2A). We also used volcano plot to view and analyze the differential expression of metabolites (Fig. 2B). The results showed metabolites M130T140 (mz: 130.1224), M538T961 (mz: 538.1734), M160T144 (mz: 160.1335), M134T49 (mz: 134.045) and M292T977 (mz: 291.9267) were discovered with most credible *P* value. According to the mass/charge ratio and *p* value of metabolites, the scatter plot was drawn to clearly see the distribution of different substances in the samples (Fig. 2C). It revealed that M130T140, M134T49, M160T144, M292T977 and M215T564 were significantly down-regulated whereas M433T331, M538T961, M537T961, M523T332 and M539T96 showed significant up-regulated expression.
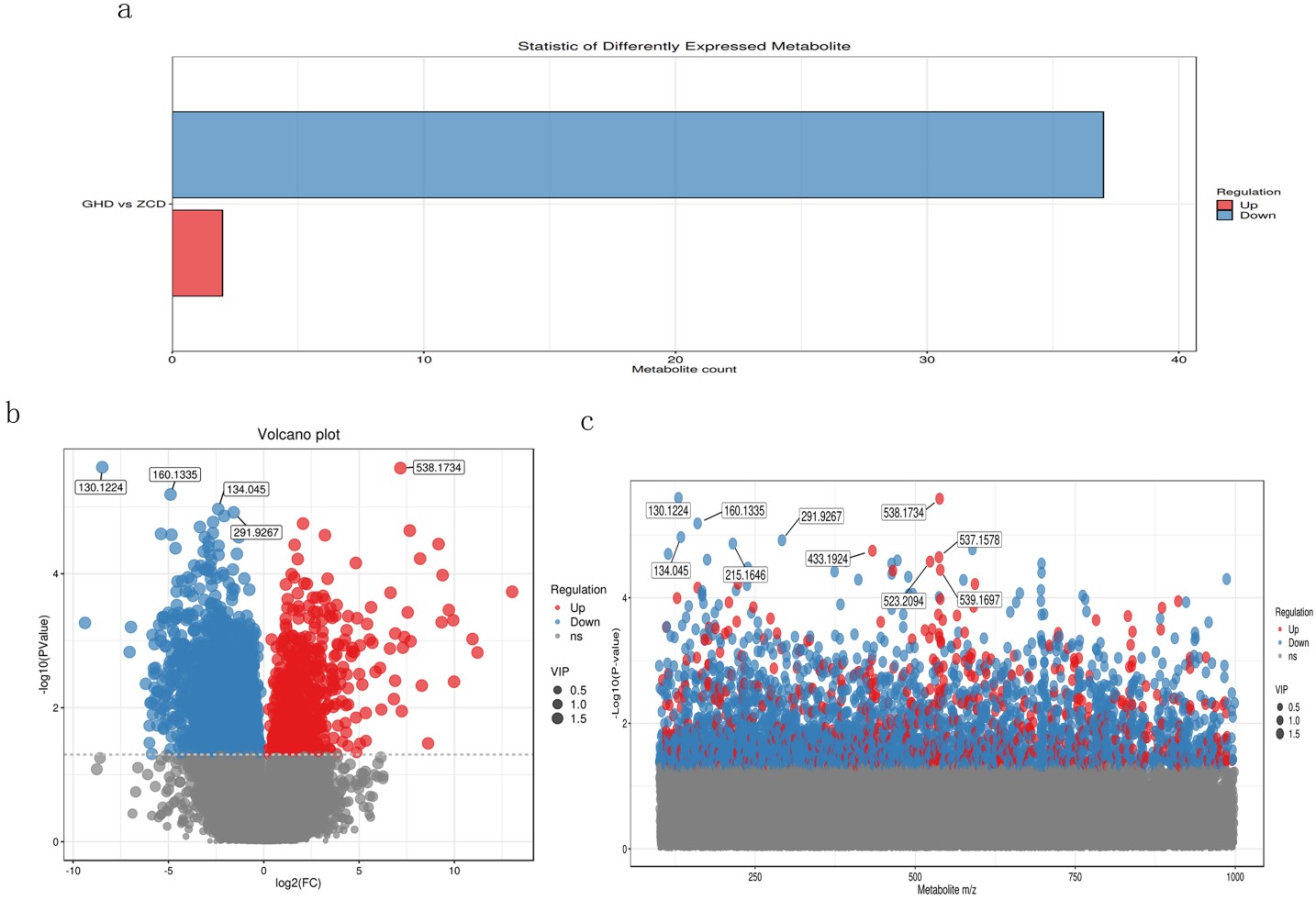

**Figure 2 Metabolomic analysis of Oriental Lily under drought stress.** (A) Statistical analysis of differential metabolites. The X axis represents the number of differential metabolites, and the Y axis represents the group comparison conditions. (B) Volcanic map of differential metabolites. Each point in the figure represents a metabolite, and the X-axis represents the logarithm value of Log2 of the multiple of quantitative difference of a metabolite in two samples. Y-axis represents the logarithm of $-\log10$ for P. (C) Scatter plot of charge ratio and $p$ value of differentially expressed metabolites. The Y-axis is the $\log^{-10}$. Red dots represent up-regulated differentially expressed metabolites, blue dots represent down-regulated differentially expressed metabolites, and gray dots represent metabolites that were detected but not expressed.

The obtained metabolites were identified by using the Human Metabolome Database (HMDB) (http://www.hmdb.ca), massbank (http://www.massbank.jp/), LipidMaps (http://www.lipidmaps.org), and mzcloud (https://www.mzcloud.org) *via* MS/MS model. Finally, 39 metabolites were obtained, among which two displayed downregulated while the other 37 were upregulated. These prominent upregulated metabolites including picolinic acid, L-Gulose, D-sorbose, UDP-D-Xylose and alpha-D-glucose. Meanwhile, an array of downregulated metabolites were also recorded such as beta-carotene and glycerophosphocholine (Figs. 3A and 3B). Correlations between different metabolites were calculated to reveal the synergy (Fig. 3C). The correlation between glycerophosphocholine and ancymidol displayed significant negative correlation, whereas positive correlation was found amongst numerous others.
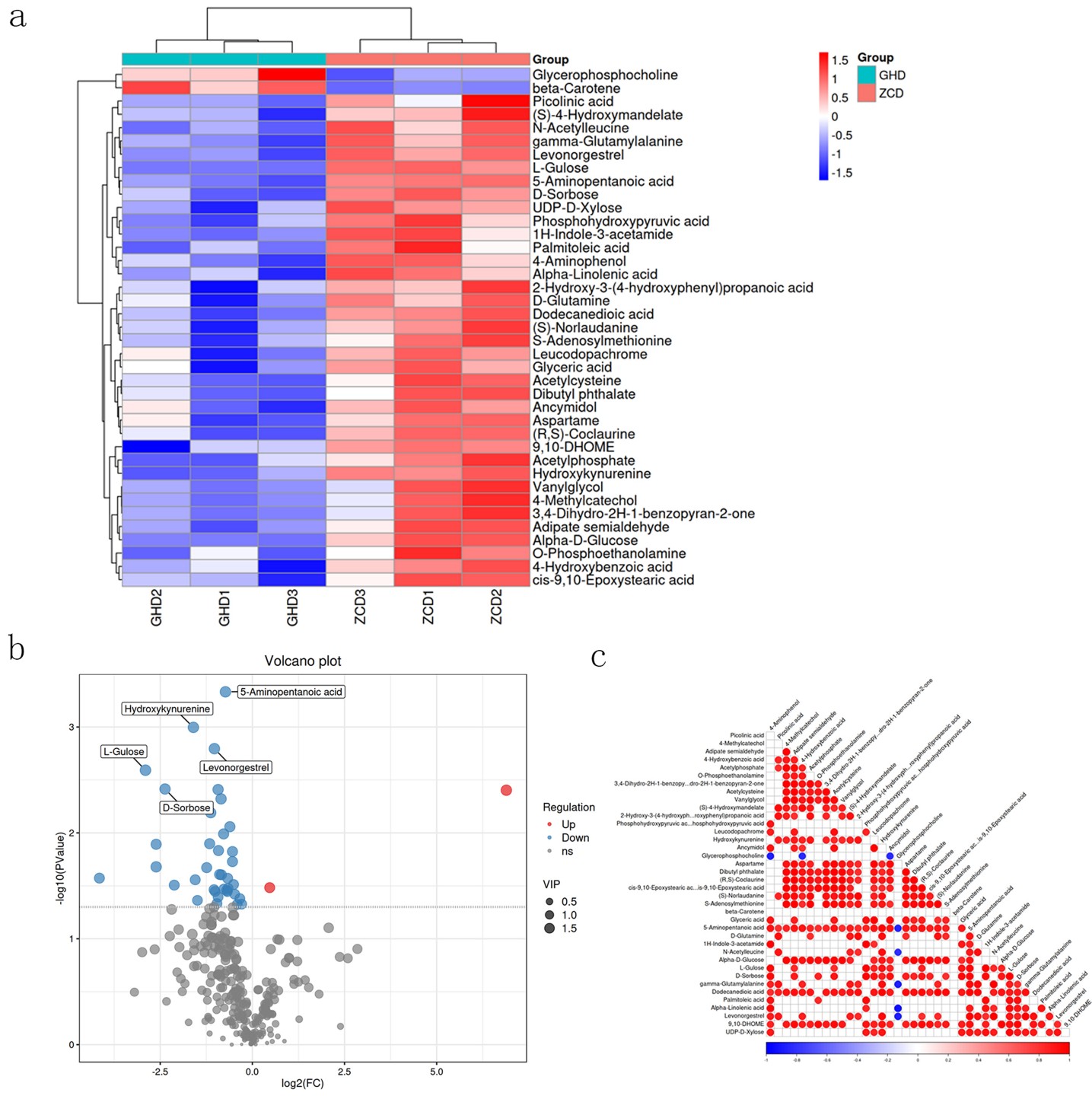

**Figure 3 Hierarchical clustering heat map of differential metabolites in Oriental Lily under drought stress.** (A) Hierarchical clustering heat map of differential metabolites. The relative content in the figure is shown by the color difference. Red color represents the higher expression level and blue color represents lower expression level. (B) Volcanic map of differentially expressed metabolites. Each point represents a metabolite, and the X-axis represents the mass/charge ratio of a metabolite. The Y-axis is the $\log^{-10}$. (C) The association heat map of differentially expressed metabolites. Both ordinate and oblique ordinate represent the names of differential metabolites, color represents correlation, red is positively correlated, blue is negatively correlated, and the darker the color, the greater the correlation.

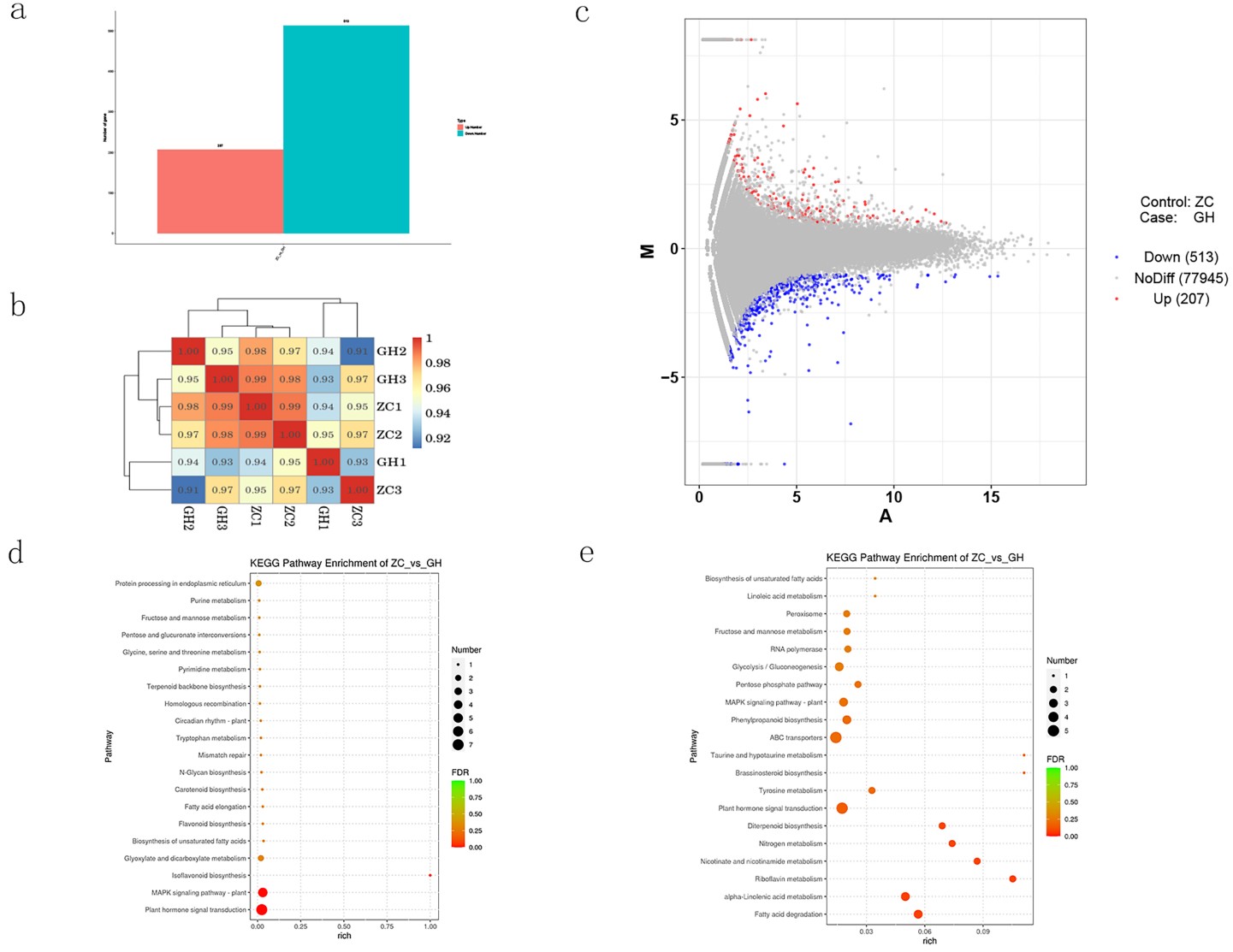

**Figure 4  Analysis of differentially expressed genes in Oriental Lily.** (A) Number of differentially expressed genes in lily, (B) correlation analysis of different samples. The different colored squares represent the high and low correlation of the two samples, (C) MA map of differentially expressed genes, (D) KEGG enrichment analysis of up-regulated genes, (E) and down-regulated genes.  

## Transcriptome analysis of Oriental Lily

The regulation of metabolites are often governed by different transcription factors (TFs). In order to understand the molecular mechanism of these metabolites in response to drought stress, the transcriptome analysis of the control sample and drought-treated sample was conducted. All unigenes were functionally annotated, mainly in databases such as NR (NCBI non-redundant protein sequences), GO (Gene Ontology), KEGG (Kyoto Encyclopedia of Genes and Genome), eggnog (evolutionary genealogy of genes: Non-supervised Orthologous Groups), Swiss-Prot and Pfam.

DESeq software was used to analyze the expression of differential genes, and the screening criteria was set as |log2FoldChange| > 1 and $p$-value < 0.05. Compared with the control, a number of 720 differentially expressed genes (DEGs) were mapped, among
which 513 were down-regulated and 207 were up-regulated (Figs. 4A–4C). KEGG enrichment analysis of DEGs (Figs. 4D and 4E) showed that up-regulated genes were mainly enriched in plant hormone signal transduction, MAPK signaling pathway, glyoxylate and dicarboxylate metabolism. On the other hand, down-regulated genes were also enriched in plant hormone signal transduction, and other different pathways including tyrosine metabolism, ABC transporters, pentose phosphate pathway, glycolysis, gluconeogenesis, RNA polymerase, fructose and mannose metabolism and peroxisome.

## Integrated analysis of transcriptome and metabolome data

To determine the correlation between DMs and DEGs, a correlation analysis was conducted based on the Pearson correlation coefficient (differential genes ($p < 0.05$) and differential metabolites (VIP > 1, $p < 0.05$)). Results with a Pearson correlation coefficient greater than 0.8 were selected. The following table shows the correlation analysis between DMs and DEGs, and it can be seen that DMs and DEGs are more negatively correlated (Fig. 5A).

A bar chart was drawn to display the significance of simultaneously enriched pathways. The results showed that the metabolic pathways potentially involved in lily drought stress response were largely associated to carbohydrate metabolism (galactose, starch and sucrose, glycolysis and gluconeogenesis) (Fig. 5B). At the same time, we selected transcripts with differential metabolites (VIP > 1, $p < 0.05$) and $p < 0.05$. Based on the metabolite information in the KEGG database, we extracted transcripts corresponding to the relevant enzymes to obtain the corresponding relationship between the two and combined them with multiple differences to display plotting (Fig. 5C). Finally, we acquired nine pairs of differentially expressed metabolites and the corresponding differential transcripts ($p < 0.05$). Interestingly, gene *TRINITY_DN2608* showed higher association with a certain metabolite and was selected for further work. The online Blast platform analysis yielded that *TRINTIY_DN2608* encode a galactose protein, mapped in the galactose pathway. Based on these findings, we put forward *TRINITY_DN2608* as a candidate gene for functional characterization.

## Identification and phenotypic observation of *TRINITY_DN2608* gene in *Arabidopsis thaliana*

The *TRINITY_DN2608* gene was heterogeneically transformed into *Arabidopsis thaliana* by using the floral dip method. The T3 generation seeds were obtained from the transgenic Arabidopsis lines carrying the *TRINITY_DN2608* gene. The seeds were disinfected with alcohol and then spread on kanamycin plate for screening and cultivation. After 14 days, *Arabidopsis thaliana* with two to four true leaves was transplanted (Fig. 6B). Leaf DNA of positive strains was extracted, and gene specific PCR primer was used to identify whether the gene was transferred into *Arabidopsis thaliana* (Fig. 6A). The transgenic plants were allowed to grow in normal conditions for 3 weeks. Following that, drought stress treatment was carried out to analyze the response of *TRINITY_DN2608* gene. In overexpressed *Arabidopsis thaliana* plants, the expression level of *TRINITY_DN2608* was 3–4 times that of the wild type (WT) (Fig. 6C). No obvious phenotypic difference was recorded under

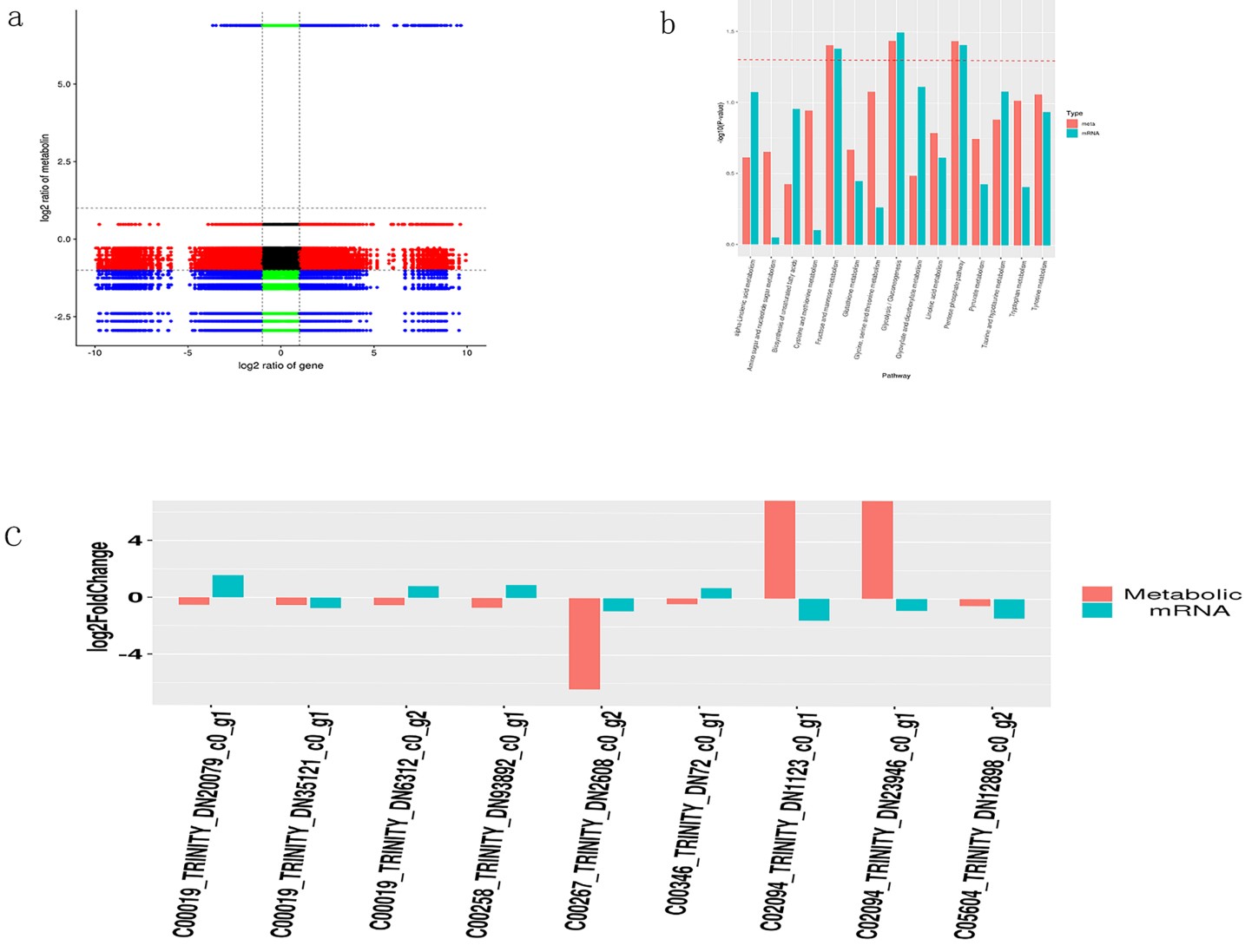

**Figure 5 Integrated transcriptome and metabolome analysis of Oriental Lily.** (A) Correlation analysis between mRNAs and metabolites. (B) Enriched P Value bar chart by KEGG, the horizontal axis represents the name of the metabolic pathway, and the vertical axis represents the *p*-values of two omics enrichment analyses. The color represents different omics. (C) Nine pairs of differentially expressed metabolites and the corresponding differential transcripts. The horizontal axis represents the names of related metabolites and transcripts, while the vertical axis represents differential expression multiples.

normal condition. However, under drought treatment, the WT plants maintained green leaves, whereas the leaves of *TRINITY_DN2608-OE* showed obvious wilting phenotype (Fig. 6D and 6E). In addition, the water loss from *TRINITY_DN2608-OE* was higher than that of the wild type between 0 and 12 h (Fig. 6F). These results suggest that allogeneic expression of *TRINITY_DN2608* gene can enhance the sensitivity of *Arabidopsis thaliana* to drought.

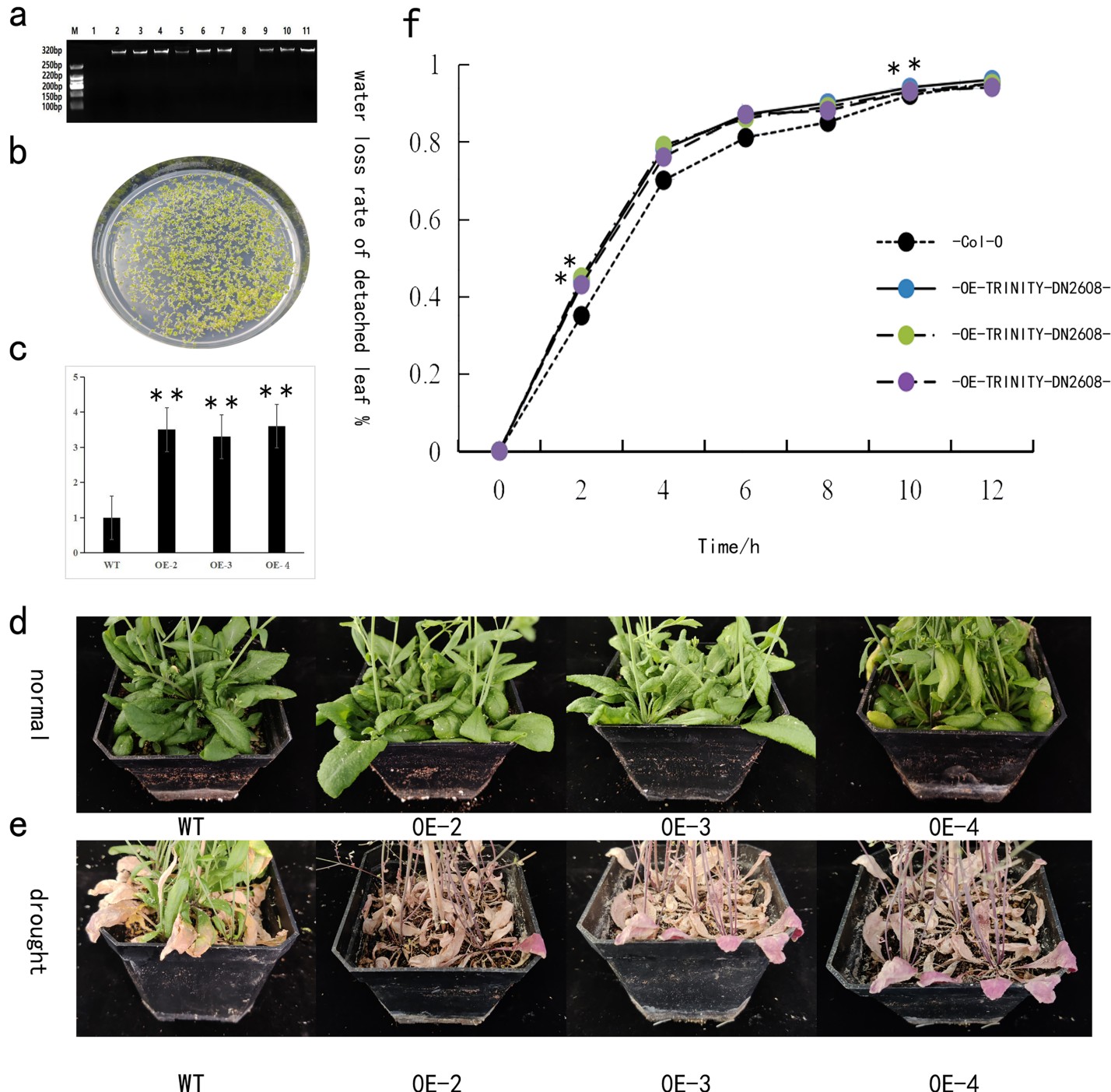

**Figure 6 Functional verification of *TRINITY_DN2608* under drought stress.** (A) Identification of the transgenic strain, (B) transgenic strain was screened on a resistant plate containing kanamycin, (C) relative expression levels of drought-related genes in transgenic and wild-type plants under drought conditions, (D) normal management of wild-type and transgenic plant phenotypes, (E) after drought treatment, wild-type and transgenic plant phenotypes, (F) statistics of *in vitro* leaf water loss rate of wild-type and transgenic plant lines. Two asterisks (**) in the figure is on the overexpression plant line, indicating a significant difference with wild-type Arabidopsis at the 0.01 level.

## DISCUSSION

Drought stress is a common, adverse factor associated with plant growth and development. Over time, natural selection and co-evolution have enabled plants that have adapted to arid environments to develop a variety of physiological, molecular and ecological drought-resistance strategies. A few studies have been conducted on the drought resistance of lily; however, these studies have mainly focused on the effects of drought stress on morphological and physiological indexes of the lily (*Hao et al., 2023*; *Driesen, De Proft & Saeys, 2023*; *Mahmood et al., 2023*). Some studies have shown that the increase of leaf thickness and Palisade cell can make the leaves transport water more efficiently and maintain the fixed morphological structure of plants (*Boughalleb, Maaloul & Abdellaoui, 2022*; *Da Cruz et al., 2023*; *Khan et al., 2023*). In line with that, our research revealed that drought stress negatively affected the leaf morphology of lily plant. Alteration in the leaf structural morphology in response to drought stress could be a vital strategy adopted by the plant to minimize the damage (*Xu et al., 2021*). Similar phenology was also observed in several other important crops such as cotton (*Dai et al., 2023*), wheat (*Raza et al., 2023*), and rice (*Zhang et al., 2023*).

A cluster of differentially expressed metabolites were mapped from the analysis. (Fig. 3A). Among them, L-Gulose, which is the precursor of ascorbic acid (vitamin C) in plant, regulating several biological and stress responsive processes (*Wolucka & Van Montagu, 2003*). D-sorbose belongs to the hexose group of rare sugars and is key in regulating plant response to osmotic damage (*Mijailovic et al., 2021*). The alpha-D-glucose, a form of glucose that is vital in regulating the key developmental processes of plant and in most cases animals (*Assefa et al., 2020*). The majority of the differentially expressed metabolites belong to the sugar metabolism pathway which led us to believe that the sugar substate could be key in plant response to drought stress (*Wang et al., 2022*; *Vicente, Annunziata & Santelia, 2022*). Drought stress has been shown to suppress the expression of genes involved in carbohydrate metabolism, including sucrose phosphate synthetase, sucrose synthetase, glyceraldehyde-3-phosphate dehydrogenase, phosphoenolpyruvate carboxylase, β-glucosidase and other related genes (*Seki et al., 2002*). Herein, three of the top ranked ones were related to sugar metabolism, namely galactose, starch and sucre, glycolysis and gluconeogenesis, where the majority of the metabolites associated with the genes were downregulated. On this basis, key genes in the aforementioned synthesis pathway were targeted for functional characterization.

The overexpression of *TRINITY DN2608* enhanced the sensitivity of lily to drought stress *via* alpha-D-Glucose. Glucose has been previously reported for enhancing plant response to drought stress by regulating water status in leaves (*Saddhe, Manuka & Penna, 2021*; *Sami et al., 2016*). In our study, we speculate that overexpression of *TRINITY DN2608* might suppress glucose level in the lily leaves thus amplifying sensitivity to drought stress.

## CONCLUSION

Joint metabolomic and transcriptomic analysis were conducted to understand the response of Oriental Lily to drought stress. The plant morphology and leaf structure of Oriental Lily

were significantly affected under drought stress. The combined analysis of transcriptome and metabolome showed that the pathway related to sugar metabolism was the main metabolic pathway of lily in response to drought stress. This study enriches and expands the response mechanism of lily to stress, providing reference and genetic resources for the cultivation of drought resistant new varieties of lily through molecular breeding methods in the future.

### Funding
This project was supported by the Fujian Province educational research project of young and middle-aged teachers grant number [JAT220593] [JAT210770]. The funders had no role in study design, data collection and analysis, decision to publish, or preparation of the manuscript.

### Grant Disclosures
The following grant information was disclosed by the authors:
Fujian Province Educational Research Project of Young and Middle-Aged Teachers Grant Number: JAT220593 and JAT210770.

### Competing Interests
The authors declare that they have no competing interests.

### Author Contributions
- Zhenkui Cui conceived and designed the experiments, authored or reviewed drafts of the article, and approved the final draft.
- Huaming Huang performed the experiments, prepared figures and/or tables, and approved the final draft.
- Tianqing Du performed the experiments, prepared figures and/or tables, and approved the final draft.
- Jianfeng Chen performed the experiments, authored or reviewed drafts of the article, and approved the final draft.
- Shuyan Huang analyzed the data, prepared figures and/or tables, and approved the final draft.
- Qushun Dai analyzed the data, authored or reviewed drafts of the article, and approved the final draft.

### Data Availability
The raw sequence data of RNA-Seq is available at NCBI: PRJNA933802.
https://www.ncbi.nlm.nih.gov/bioproject/933802.
The plant measurements are available in the Supplemental File.

## Supplemental Information

Supplemental information for this article can be found online at http://dx.doi.org/10.7717/peerj.16658#supplemental-information.

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
