# Peer review of "Integrated transcriptome and metabolome revealed the drought responsive metabolic pathways in Oriental Lily (Lilium L.)"

_PeerJ, doi:10.7717/peerj.16658_

## Round 0.1 · original submission · Major Revisions

The manuscript has been reviewed by three reviewers and I must admit that I agree in many respects with the points addressed by reviewer #2: The abstract does not provide enough information about the scientific question, and fails to highlight main results in an understandable manner. The summary is too superficial without correlating transcriptomic to metabolite data. Regarding the analysis of omics data, it seems that you refer to very huge annotation bins ("glycolic pathways") e.g. without differentiating between ana- or catabolism. The interpretations made from such analysis remain therefore too vague to justify their conclusion. Moreover, It is unclear whether you refer to genes or metabolites (see e.g. the sentence "Interestingly, a target gene named TRINITY_DN2608 (encoding a type of Alpha-D-Glucose) was gained and transformed..."). Overall, I ask you to carefully revise and maybe rewrite large parts of the manuscript following the advice from all reviewers.

**Language Note:** The review process has identified that the English language must be improved. PeerJ can provide language editing services - please contact us at copyediting@peerj.com for pricing (be sure to provide your manuscript number and title). Alternatively, you should make your own arrangements to improve the language quality and provide details in your response letter. – PeerJ Staff

Reviewer 1 ·

Basic reporting

The study on Oriental Lily under drought stress using both transcriptomic and metabolomic approaches provides comprehensive insights into the molecular mechanisms behind the drought response. The integration of multi-omics data, functional validation of a candidate gene in Arabidopsis, and a combination of phenotypic, anatomical, and molecular analyses offers a profound and holistic understanding of the drought resistance mechanisms in Lily.

1) The manuscript could benefit from a more concise presentation, especially in the results section, to improve clarity and readability.

2) Statistical analyses were missing in Figure 1c,d,e,f (particularly for the 20-day group) and Figure 6I. In the context of Figure 6I, there was no noticeable difference in the water loss rate of detached leaves across all groups at the 12h mark. This point needs addressing. Do the authors attribute this to experimental error, or does it have a biological implication?

Experimental design

1) While a significant number of metabolites have been identified, a deeper interpretation of their roles in the drought response, beyond just upregulation or downregulation, would be beneficial. An exploration of their specific biological roles and interactions would provide a richer understanding of their function.

2) The functional implications of the reduced glucose metabolism level in lily under drought stress could be elaborated more. The authors should offer mechanistic insights into how changes in this pathway might influence drought tolerance could enhance the study's impact. Does the reduction in glucose metabolism offer any protective mechanism against drought stress? How do these metabolic shifts interact with other pathways to shape the overall drought response?

Validity of the findings

no comment

Additional comments

no comment

Reviewer 2 ·

Basic reporting

.

Experimental design

.

Validity of the findings

.

Additional comments

The writing and description of this article are very confusing, listing many unnecessary and unclear formulas. The images and text descriptions are unclear and not enough in results, such as what’s the relationship between metabolites and unigenes in Figure 5? Why select them to specifically shown in figure 5c? What does DN2608 encoded Galactose mean? The abbreviations and full names in the text do not correspond. It’s unbelievable that the authors obtain 37285 products in metabolic anaylysis? What method do they perform? There are a lot of repetitions in the material methods and the description of the results in the article, and the analysis and discussion of the results are not in-depth enough. There are a large number of writing errors in the article, including incomplete sentences, grammar errors, and etc.The pixel size of the image is extremely low, and there is a huge difference in the size of the text in the image.
The research topic in this article is interesting, but the experimental methods and results display have not reached the level that can be published at all. It is strongly recommended that the author spend more time revising and improving before resubmitting to any journal.

Reviewer 3 ·

Basic reporting

'no comment'

Experimental design

'no comment'

Validity of the findings

'no comment'

Additional comments

Lily is an important economic ornamental plant. The authors used the transcriptome and metabolome analysis to find some key genes involved in the drought tolerance. The results of this manuscript is very meaningful, however, there are some comments should be addressed.

1. The author should provide key message in the “Abstract”, and improve the language quality.
2. In line 31 and 32, the gene names should not use bold type, the author should be learn more about the guidance of the PeerJ journal.
3. The author should reconsider the keywords in this manuscript.
4. In the “Materials and Methods”, the detail of drought treatment should be provided.
5. In the “2.4 Sample preparation for metabolome analysis”, this part should be rewritten.
6. For the part 2.7 and 2.8, I think the two parts should be integrated together. And the author should cite references for the expression calculation.
7. In 2.9, for the relative expression calculation, the author should cite this reference, there is no need list the calculation processes. The reference as follow:
Livak, K. J., & Schmittgen, T. D. (2001). Analysis of relative gene expression data using real-time quantitative PCR and the 2− ΔΔCT method. methods, 25(4), 402-408.
8. In 226, the “EcoR I and Not I.” should be used Italy font.
9. In 230, “OD600 value” should be OD600.
10. The quality of all the figures should be improved in this manuscript.
11. In 353 to 255, the formula of water loss rate should be in the “Materials and Methods” part.
12. The language quality of this manuscript should be improved, it is very important for the readers to understand your key message.

---

## Round 0.2 · accepted · Accept

Your manuscript has been reviewed again by one of your previous reviewers and acknowledges that you addressed all comments and suggestions to further improve your manuscript properly.

Reviewer 1 ·

Basic reporting

The authors have addressed all comments properly, and I do not have any further suggestions.

Experimental design

n/a

Validity of the findings

n/a

Additional comments

n/a